# Don't Waste a Single Annotation: Improving Single-Label Classifiers Through Soft Labels

**Ben Wu, Yue Li, Yida Mu, Carolina Scarton, Kalina Bontcheva** and **Xingyi Song**

Department of Computer Science, The University of Sheffield, Sheffield, UK

bpwu1@sheffield.ac.uk

## Abstract

This paper addresses the limitations of the common data annotation and training methods for objective single-label classification tasks. Typically, in such tasks annotators are only asked to provide a single label for each sample and annotator disagreement is discarded when a final hard label is decided through majority voting. We challenge this traditional approach, acknowledging that determining the appropriate label can be difficult due to the ambiguity and lack of context in the data samples. Rather than discarding the information from such ambiguous annotations, our soft label method makes use of them for training. Our findings indicate that additional annotator information, such as confidence, secondary label and disagreement, can be used to effectively generate soft labels. Training classifiers with these soft labels then leads to improved performance and calibration on the hard label test set.

## 1 Introduction

Reliable, human-annotated data is crucial for training and evaluation of classification models, with the quality of annotations directly impacting the models' classification performance. Traditionally, in order to ensure high quality annotated data, multiple annotators are asked to judge each individual data instance, and the final 'gold standard' *hard label* is determined by majority vote.

However, this hard label approach tends to ignore valuable information from the annotation process, failing to capture the uncertainties and intricacies in real-world data (Uma et al., 2021). An emerging alternative approach that addresses these limitations is the use of soft labels through techniques such as Knowledge Distillation (Hinton et al., 2015), Label Smoothing (Szegedy et al., 2016), Confidence-based Labeling (Collins et al., 2022), and Annotation Aggregation (Uma et al., 2020). These soft label approaches demonstrate potential for improved robustness (Peterson et al.,

2019), superior calibration, enhanced performance (Fornaciari et al., 2021) and even enable less than one-shot learning (Sucholutsky and Schonlau, 2021).

This paper's primary focus is on exploring effective ways for improving classification performance using soft labels. Our experimental findings indicate that confidence-based labelling significantly enhances model performance. Nevertheless, the interpretation of confidence scores can also profoundly influence model capability. Given the variability in confidence levels among different annotators (Lichtenstein and Fischhoff, 1977), aligning these disparate confidence levels emerges as the central research question of this paper.

To address this challenge, we propose a novel method for generating enhanced soft labels by leveraging annotator agreement to align confidence levels. Our contributions include:

- We demonstrate how classification performance can be improved by using soft labels generated from annotator confidence and secondary labels. This presents a solution to the challenge of generating high-quality soft labels with limited annotator resources.
- We propose a Bayesian approach to leveraging annotator agreement as a way of aligning individual annotators' confidence scores.
- We introduce a novel dataset to facilitate research on the use of soft labels in Natural Language Processing.[1]

## 2 Related Work

Current research typically interprets annotator disagreement in two primary ways, either by capturing diverse beliefs among annotators, or by assuming a single ground truth label exists despite disagreement (Rottger et al., 2022; Uma et al., 2021). This paper focuses on situations where the latter

---

[1]Dataset can be found at: https://github.com/GateNLP/dont-waste-single-annotation

viewpoint is more applicable. Thus, we focus on traditional "hard" evaluation metrics such as F1-score which rely on a gold-label, despite the emergence of alternative, non-aggregated evaluation approaches (Basile et al., 2021; Baan et al., 2022; Basile et al., 2020). This is made possible because we evaluate on high-agreement test sets, where the 'true' label is fairly certain.

Aggregation of annotator disagreement generally falls into two categories: aggregating labels into a one-hot *hard label* (Dawid and Skene, 1979; Hovy et al., 2013; Jamison and Gurevych, 2015; Beigman and Beigman Klebanov, 2009), or modeling disagreement as a probability distribution with *soft labels* (Sheng et al., 2008; Uma et al., 2020; Peterson et al., 2019; Davani et al., 2022; Rodrigues and Pereira, 2018; Fornaciari et al., 2021).

Similar to Collins et al. (2022), our study explores how soft labels can be generated from a small pool of annotators, using additional information such as their self-reported confidence. This has benefits over traditional hard/soft label aggregation, which requires extensive annotator resources and/or reliance on potentially unreliable crowd-sourced annotators (Snow et al., 2008; Dumitrache et al., 2018; Poesio et al., 2019; Nie et al., 2020).

# 3 Methodology

In order to generate soft labels, our methodology requires annotators to provide confidence scores. Figure 1 shows how each annotator provides both a primary class label and a confidence rating that represents their certainty. This ranges from 0 to 1, with 1 representing 100% confidence.[2] In addition, annotators can also provide an optional 'secondary' class label. This represents their selection of the second most probable class. Thus, formally, the annotation of the text $x_i$ by an annotator $a_m$ consists of a primary label $l_{im}$, its confidence rating $c_{im}$, and an optional secondary label $l_{im}^2$. We use $y_i$ to denote the text's true label.

Overall, there are three steps to generating soft labels as shown in Figure 1:

1. Annotator confidences are calibrated using our Bayesian method (Section 3.1)
2. Annotations are converted to soft labels (Section 3.2)
3. Annotator soft labels are merged into a final

---

[2]In practice, annotators can provide this directly as a percentage or choose from a Likert-style numerical rating (e.g. 1-5) that is then converted to 0-1 scale.

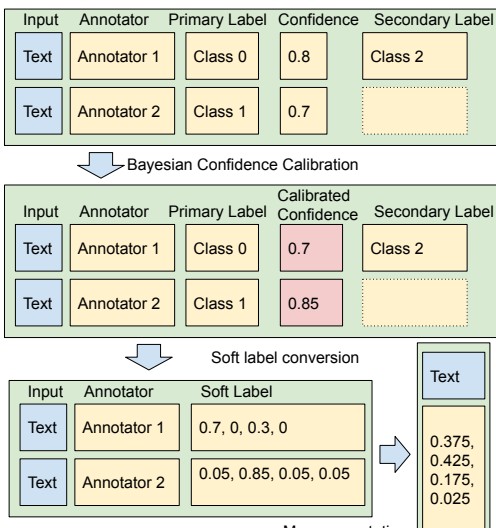

Figure 1: Soft label conversion pipeline

soft label (Section 3.2)

## 3.1 Bayesian Confidence Calibration

To calibrate confidence levels across annotators, we use annotator agreement as a proxy for reliability. For each annotator, we consider the level of agreement obtained (across all their annotations) when they have expressed a particular confidence score, and use this to re-weight their confidence. The process comprises two steps:

First, we compute the probability of the primary label ($l_{im}$) according to the confidence level ($c_{im}$). This step is agnostic to the identity of the annotator.

$$P(\hat{y}_i = l_{im}|c_{im}) = \frac{P(c_{im}|l_{im})P(l_{im})}{P(c_{im})} \quad (1)$$

Where $P(l_{im})$ is the prior, $P(c_{im}|l_{im})$ is the likelihood of the confidence score assigned to the primary label and $P(c_{im}) = P(c_{im}|l_{im})P(l_{im}) + P(c_{im}|\neg l_{im})P(\neg l_{im})$ is the marginal probability. In this paper, we just make a simple assumption $P(l_{im}) = 1/C$ and $P(\neg l_{im}) = (C-1)/C$, where $C$ is the total number of possible classes. We also assume that $P(c_{im}|y_i) = c_{im}$ and $P(c_{im}|\neg y_i) = 1 - c_{im}$.

In the second step, using information about agreement, we compute the calibrated probability of the primary label ($l_{im}$) given the specifc annotator ($a_m$).

$$P(y_i = l_{im}|a_m) = \frac{P(a_m|l_{im})P(l_{im})}{P(a_m)} \quad (2)$$

Where our updated prior $P(l_{im}) = P(\hat{y}_i)$ calculated from Equation 1. $P(a_m|l_{im})$ is the likelihood that an annotator assigns the label $l_{im}$ matching the true label, $y_i$, and $P(a_m) = P(a_m|y_i)P(\hat{y}_i) + P(a_i|\neg y_i)P(\neg\hat{y}_i)$.

We calculate this likelihood based on annotator disagreement:

$$P(a_m|y_i) = \frac{Count(a_m \cap \neg a_m, a_m, li)}{Count(a_m \cap \neg a_m, \neg a_m, li)} \quad (3)$$

$Count(a_m \cap \neg a_m, a_m, li)$ is the number of samples that involves annotator $a_m$ and any other annotator(s) $\neg a_m$, where both $a_m$ and at least one other annotator provided the label $li$. $Count(a_m \cap \neg a_m, \neg a_m, li)$ is the number of samples that involves annotator $a_m$ and other annotator(s) $\neg a_m$, where at least one annotator provided the label $li$.

Similarly, we calculate:

$$P(a_m|\neg y_i) = \frac{Count(a_m \cup \neg a_m, a_m, li)}{Count(a_m \cap \neg a_m, \neg a_m, \neg li)} \quad (4)$$

$Count(a_m \cup \neg a_m, a_m, li)$ is the number of samples that involves annotator $a_m$ and other annotator(s) $\neg a_m$ where $a_m$ annotated the sample as $li$ but others did not. $Count(a_m \cap \neg a_m, \neg a_m, \neg li)$ is the number of samples that involves annotator $a_m$ and other annotator $\neg a_m$ where $\neg a_m$ did not annotate the sample as label $li$.

We also employ fallback mechanisms to mitigate cases where annotator agreement cannot be reliably estimated. If the number of counts ($< 3$ in our experiments) is insufficient to calculate posterior probability, we fall back to using the prior confidence score.

## 3.2 Soft Label Conversion & Merging Annotations

Once we have calibrated confidence, $P(y_i|a_m)$, we assign this probability to the primary class $l_i$, and assign $1 - P(y_i|a_m)$ to the secondary class $l_i^2$. Thus, in Figure 1, Annotator 1's soft label consists of 0.7 for their primary class and 0.3 for their secondary class. If a secondary class label is not provided by the annotator, we just uniformly distribute the confidence level to the other classes. In Annotator 2's case, this means 0.85 for their primary class and 0.05 for the three remaining classes.

Once we have generated soft labels for each annotator, we merge these into a final soft label by taking the mean of each class.

## 4 Experiments

### 4.1 Datasets

We experiment with two datasets: VaxxHesitancy (Mu et al., 2023) and the COVID-19 Disinformation Corpus (CDS) (Song et al., 2021). Both datasets release the confidence scores which annotators provided alongside their class labels (annotators are denoted by a corresponding anonymous ID).

#### 4.1.1 COVID-19 Disinformation Corpus (CDS)

CDS (Song et al., 2021) includes 1,480 debunks of COVID-19-related disinformation from various countries. The debunks are classified into ten topic categories (e.g., public authority, conspiracies and prominent actors). The number of annotators per instance ranges from one to six. Each annotator has provided only one first-choice topic class and their confidence score for each annotated debunk ($0 \leq c_{im} \leq 9$).

#### 4.1.2 VaxxHesitancy

VaxxHesitancy (Mu et al., 2023) consists of 3,221 tweets annotated for stance towards the COVID-19 vaccine. Each instance is categorised into pro-vaccine, anti-vaccine, vaccine-hesitant, or irrelevant. The number of annotators per tweet ranges from one to three. Annotators provide a first-choice stance category and a confidence score ($1 \leq c_{im} \leq 5$).

**VaxxHesitancy Additional Annotation** As our aim is to investigate how additional information provided by annotators could impact classification performance, we also explore the integration of a secondary label for instances where annotators have expressed uncertainty about their primary label choice.

As none of the original datasets had such secondary labels, we undertake an additional round of data annotation, based on the original annotation guidelines. We introduce two new tasks in this data annotation round: 1) For all instances (train + test set) exhibiting low confidence (less than 4), we optionally request that annotators provide a 'second stance' label. We guide annotators to propose this if they believe it to be appropriate, even if it wasn't their primary choice. Consequently, we add 569 additional second-choice stances. 2) We assign a third annotator to all instances annotated by two annotators. As a result, we obtain a majority vote for the majority of annotated tweets. This majority vote can be employed for hard label training in subsequent experiments.

| | VaxxHesitancy (5 fold) | | | CDS (5 fold) | | |
|---|---|---|---|---|---|---|
| | Labels Used | F1 Macro | ECE Calibration | Labels Used | F1 Macro | ECE Calibration |
| **Test-set only** | | | | | | |
| Hard Label | 1 and 2 | 66.9 | 0.262 | 1 | 65.2 | 0.182 |
| Soft Label | 1 and 2 | **68.7** | **0.192** | 1 | **67.7** | **0.161** |
| **Train+Test-set** | | | | | | |
| Without confidence | | | | | | |
| Hard Label | 1 and 2 | 70.2 | 0.237 | 1 | 67.1 | 0.211 |
| Dawid Soft Label | 1 | **71.7** | **0.162** | 1 | 19.6 | - |
| Soft Label (primary label) | 1 | 71.0 | 0.171 | 1 | **68.2** | **0.160** |
| Soft Label (primary and secondary) | 1 and 2 | 70.7 | 0.188 | - | - | - |
| With confidence | | | | | | |
| Hard Label | 1 and 2 | 73.1 | 0.198 | 1 | 69.0 | 0.190 |
| Hard Label (label smoothed 0.1) | 1 and 2 | 68.8 | 0.167 | 1 | 68.8 | **0.096** |
| Soft Label (from primary label) | 1 | 73.7 | 0.107 | 1 | **71.1** | **0.096** |
| Soft Label (primary and secondary) | 1 and 2 | **74.5** | **0.106** | - | - | - |
| With confidence + bayes calibration | | | | | | |
| Bayesian Soft Label | 1 and 2 | **75.2** | **0.099** | 1 | 70.4 | 0.118 |

Table 1: Evaluation results for the CDS and VaxxHesitancy datasets. The 'Labels Used' column indicates whether the hard/soft labels are generated using only the primary label (1), or considering the secondary label as well (2). For the CDS dataset, all labels are generated using only primary since no secondary labels are available.

### 4.1.3 Dataset Split

We construct the test sets to contain instances where annotators reach agreement on every instance with high confidence scores. For VaxxHesitancy, we follow the original train-test split by including instances whose confidence scores are larger than three in the test set. For CDS, the test set has debunks that are labelled by more than one annotator with confidence scores larger than six (on their original 10 point scale). Given the limited size of this subset, data with only one annotation but very high confidence scores is also included in the CDS test set. Summary of the statistics is in the Appendix Table 2.

### 4.2 Baselines & Ablations

We compare our methods against a variety of hard/soft label aggregation strategies, which make use of annotator confidences/secondary labels to varying degrees.

**Hard label w/o confidence** We employ majority voting for hard labels. In the absence of consensus, a class category is chosen at random.

**Hard label with confidence:** For each $x_i$, we aggregate $a_i$ to estimate a single hard label $\hat{y}_i$ by giving different weights to $l_{im}$ according to the annotator confidence $c_{im}$.

**Dawid-Skene Soft label:** We utilise an enhanced Dawid-Skene model (Passonneau and Carpenter, 2014) as an alternative to majority voting, and use the model's confusion matrices to generate soft labels. This model only relies on class labels does not make use of additional information.

**Label Smoothing Soft Label:** For each class, we use a mixture of its one-hot hard label vector and the uniform prior distribution over this class (Szegedy et al., 2016).

**Soft label w/o annotator confidence** We explore generating soft labels using only annotator disagreement. In this approach, we assign 0.7 to the primary stance, 0.3 to the secondary stance label (if available), or evenly distribute the remaining probability among all other classes.

### 4.3 Experimental Setup

We conduct 5-fold cross-validation where each fold contains the entire training set, and 4/5 folds of the test set. We also investigate a second scenario in which we perform 5-fold cross-validation only on the test set. These two scenarios allow us to investigate the performance of soft labels when the level of annotator agreement differs.

These two scenarios are motivated by the fact that the train-test splits of our datasets contain an uneven distribution of samples: low annotator agreement samples were placed in the train set and high-agreement samples in the test set. This is necessary to enable evaluation against a gold-standard test set. However, for generating soft labels, we want to use a mixture of high-agreement and low-agreement annotations, so we include a portion of the original test set for training.

We perform experiments using Pytorch (Paszke et al., 2019) and the Transformers library from HuggingFace (Wolf et al., 2020). We fine-tune COVID-Twitter BERT V2, a BERT large uncased model that has been pre-trained on COVID-19 data (Müller et al., 2023). We fine-tune for 20 epochs with learning rate as 2e-5 (1 epoch warm-up followed by linear decay) and batch size as 8, with

AdamW optimizer (Loshchilov and Hutter, 2017). We use cross-entropy loss. The model performance is evaluated with macro-F1 due to the imbalanced datasets, and we use expected calibration error (ECE) (Naeini et al., 2015) to measure model calibration.

For both the VaxxHesitancy and CDS datasets, we harmonise the range of confidence scores. In the case of the VaxxHesitancy dataset, this involves converting a confidence score of 5 to 1, 4 to 0.9, and so forth, down to 1 being converted to 0.6. Similarly, for the CDS dataset, a confidence score of 10 is converted to 1, 9 to 0.95, and so on, with 1 also becoming 0.6. [3]

## 5 Results

**Soft labels improve classification performance across all of our scenarios.** Table 1 presents the results on the VaxxHesitancy and CDS datasets. Soft labels surpass hard labels for both datasets, with and without confidence, as well as for the test-only and train + test set scenarios. In the case of test-set only, soft labels achieve 68.7 F1 Macro vs hard label's 66.9 (VaxxHesitancy) and 67.7 vs 65.2 (CDS). As previously mentioned, the test set comprises of only high-agreement samples, so this indicates that soft labels are beneficial for learning even when there is not a lot of disagreement between annotators and they are relatively certain.

**Combining confidence scores and secondary labels generates better soft labels.** Using annotators' self-reported confidence scores helps to improve soft labels, as shown by the F1 and calibration improvements between soft labels in the 'with' and 'without confidence' settings (Table 1). Alternative approaches such as Dawid Skene are able to outperform soft labels when confidence scores are not available (71.7 vs 71.0 on VaxxHesitancy). However, once confidence information is introduced, soft labels significantly improves and outperforms alternatives, achieving 73.7 (from 71.0) on VaxxHesitancy and 71.1 (from 68.2 on CDS).

Furthermore, for the VaxxHesitancy dataset, once secondary label information is included, classification performance is further improved from 73.7 to 74.5. This suggests that more consideration should be taken during the data annotation stage to collect such information, as it can be greatly beneficial for the creation of effective soft labels.

**Bayesian calibration outperforms other methods on the VaxxHesitancy dataset.** By incorporating the full annotation information, i.e., confidence, secondary label, and annotator agreement, our Bayesian soft label method achieves a 75.2 F1 Macro score on the VaxxHesitancy dataset, improving upon 74.5 from the soft label stance 1 and 2. In addition, Bayesian soft label also has the best confidence alignment (ECE) score.

However, on the CDS dataset, despite outperforming hard labels, our Bayesian method fails to improve upon soft labels. Its adjustments to soft labels results in a fall from 71.1 to 70.4. We believe this is due to the characteristics of the CDS dataset, which has 10 possible classes (vs the 4 of VaxxHesitancy), an increased range of possible confidences (1-9), as well as fewer overall samples. Because there is less annotator overlap, this greater range of options makes it more difficult to accurately estimate annotator reliability on a per-confidence score level. This reveals a direction in which our Bayes method could be improved, as it is currently reliant on sufficient overlap between an individual annotator and their peers.

## 6 Conclusion

We demonstrate the benefits of using soft labels over traditional hard labels in classification tasks. We propose a novel Bayesian method for annotation confidence calibration, and efficiently utilising all available annotation information, outperforming other methods for the VaxxHesitancy dataset. The performance improvements offered by soft labels suggests the importance of collecting additional information from annotators during the annotation process, with annotator confidence being particularly important.

## Acknowledgments

This work has been co-funded by the UK's innovation agency (Innovate UK) grant 10039055 (approved under the Horizon Europe Programme as vera.ai, EU grant agreement 101070093).[4], the European Union under action number 2020-EU-IA-0282 and agreement number INEA/CEF/ICT/A2020/2381686 (EDMO Ire-

---

[3]We manually tested different confidence conversion scales and this conversion yields the best classification performance. See Appendix 3 for an alternative conversion strategy.

---

[4]https://www.veraai.eu/

land).[5]. Ben Wu is supported by an EPSRC Doctoral Training Partnership Grant and Yue Li is supported by a Sheffield–China Scholarships Council PhD Scholarship.

## Limitations

Our dataset construction introduces secondary labels that are not provided by the same annotators as those who created the original dataset, which may not accurately reflect the choices that the original annotators would have made.

As discussed in the Results section, the CDS dataset was more challenging due number of class labels and number of samples. This caused the Dawid Skene model to perform poorly. This issue may be alleviated using Laplace Smoothing, but we did not explore this due to time constraints.

Another important limitation of our Bayesian method is its assumption that an individual annotator's level of agreement with their peers is a good proxy for their reliability. This leaves it vulnerable to situations where there is high agreement between poor annotators.

Even though our soft label method is effective, it is not compared against 'traditional' soft labels, which are constructed by aggregating many annotator labels per sample, since this would necessitate the large-scale annotation of the two datasets by many users, which we are trying to avoid as our goal is to reduce the amount of annotators and annotation effort required.

Finally, we observed high variance across folds during cross-validation. We believe this was due to the small size of test set as well as the presence of 'hard-to-classify' samples in certain folds. These were samples where annotators relied on multimodal information to come to a decision (e.g. viewing a video embedded in the tweet). Our model is only provided with text, and so struggles on these samples.

## Ethics Statement

Our work has received ethical approval from the Ethics Committee of our university and complies with the research policies of Twitter and follows established protocols for the data annotation process. The human annotators were recruited and trained following our university's ethics protocols, including provision of an information sheet, a consent form, and the ability to withdraw from annotation at any time. For quality and ethics reasons, volunteer annotators were recruited from our university, rather than via Mechanical Turk.

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

## A  Appendix

Table 2 shows information about the composition of the two datasets used in our experiments.

|  | VaxxHesitancy | | CDS | |
| --- | --- | --- | --- | --- |
|  | Train | Test | Train | Test |
| Number of items | 2,790 | 431 | 965 | 515 |
| Avg annotations per item | 1.25 | 2 | 1.38 | 1.58 |
| Avg confidence score | 4.16 | 4.63 | 6.73 | 8.53 |

Table 2: Summary of the datasets

Table 3 shows the effect of changing the annotator confidence conversion scale from the one presented in the main section of this paper (9: 1.0 ... 1:0.6) to an alternative one (9: 1.0, 8: 0.9 ... 1: 0.1). By comparing between the two columns, we can see that this change leads to a drop of  1 F1 point for the resulting soft labels. This highlights the importance of selecting a good initial conversion scale.

| | Covid Misinfo (5 fold) | | |
| --- | --- | --- | --- |
| | | 1-0.1 conversion | 1-0.6 conversion |
| | **Uses confidence** | **F1 Macro** | **F1 Macro** |
| Experiments on test-set only | | | |
| Hard Label | yes | 65.18 | - (same as left) |
| Soft Label | yes | **66.51** | **67.71** |
| Experiments on train + test-set | | | |
| Hard Label (majority vote, ties broken by confidence) | yes | 68.53 | 69.02 |
| Soft Label | yes | **70.59** | **71.11** |
| Experiments with no confidence | | | |
| Hard Label | no | 67.13 | - (same as left) |
| Soft Label | no | **68.15** | - (same as left) |
| Dawid Soft Label | no | 19.58 | - (same as left) |
| Experiments with confidence | | | |
| Hard Label | yes | 68.53 | 69.02 |
| Hard Label (label smoothed 0.1) | yes | 68.69 | 68.75 |
| Hard Label (label smoothed 0.3) | yes | 68.86 | 68.44 |
| Soft Label | yes | **70.59** | **71.11** |
| Experiments with bayesian calibration | | | |
| Soft label with Bayesian | yes | 69.75 | 70.40 |

Table 3: CDS Results with different confidence conversions.