# OpenReview forum: "Don't waste a single annotation: improving single-label classifiers through soft labels"
_EMNLP/2023/Conference — EMNLP 2023 Findings_

### Official Review · Reviewer_qiyT · 2023-08-05

**Soundness:** 4

**Excitement:**

3: Ambivalent: It has merits (e.g., it reports state-of-the-art results, the idea is nice), but there are key weaknesses (e.g., it describes incremental work), and it can significantly benefit from another round of revision. However, I won't object to accepting it if my co-reviewers champion it.

**Missing References:**

[1] Zhang et al. 2021. WRENCH: A Comprehensive Benchmark for Weak Supervision.

**Paper Topic And Main Contributions:**

The paper proposes a new method to improve the annotations by replacing the hard labels with soft ones generated using additional annotator information (e.g., confidence, disagreement, and the secondary label).

**Questions For The Authors:**

- Some notation is unclear. E.g., Eq. 1 & 2: is it assumed that we calculate the probability of _estimated_ y (i.e., y hat) in Eq. 1 and the probability of _true_ y in Eq. 2? If so, why in line 134 is there the probability of _estimated_ y (i.e., y hat) again?
- The experimental setup is unclear to me. As far as I get it from Section 4, 4/5 of the test set was included in the training set. Does it mean the test set was involved in the training?

+ see above.

**Reasons To Accept:**

- Annotation improvement is an interesting and important task which could be relevant in many domains.
- The method is well motivated and theoretically sound.
- The article is well written and easy to follow.

**Reasons To Reject:**

- The method is compared towards simple and old baselines. More recent baselines are needed (e.g., weakly supervised baselines, among which there are methods that use soft labels as well, see [1]).
- All datasets used in the experiments are small. There is no guarantee that the method performs well on larger datasets.
- More result analysis is needed.
- Line 335: I couldn't find these numbers in Table 1.
- Line 361: If the authors observed high variance in runs, this is worth entering in Table 1 and discussing.
- No hyperparameter search space is provided. Also, not clear how the best values were retrieved given that there is no development set.

[1] Zhang et al. 2021. WRENCH: A Comprehensive Benchmark for Weak Supervision.

**Reproducibility:**

4: Could mostly reproduce the results, but there may be some variation because of sample variance or minor variations in their interpretation of the protocol or method.

**Reviewer Confidence:**

5: Positive that my evaluation is correct. I read the paper very carefully and I am very familiar with related work.

**Typos Grammar Style And Presentation Improvements:**

- Line 151: typo
- Line 163: typo

---

> ### Author Rebuttal · Authors · 2023-08-28
>
> Thank you for your thorough feedback and review!
>
> **1. Weak Supervision Baselines**
>
> We agree that there are many exciting weak supervision techniques that could also be used as benchmarks. However, we highlight that the Majority Vote baseline we use remains very competitive for our task. In the WRENCH paper you reference, the authors recommend ‘Majority Voting’ and ‘MeTaL’ as the two best-performing weak supervision methods for classification. In Table 12, which focuses on textual classification with Bert/Roberta (similar to our task scenario), Majority Voting (both hard and soft) outperforms MeTaL labels (both hard and soft). This suggests that MV is an appropriate baseline.
>
> **2. Dataset size**
>
> Dataset size is definitely a limitation of our findings. However, as discussed in our response to Reviewer MMN4, we were limited by the lack of publicly available datasets that offered individual annotator confidence scores. This lack is also what motivated our additional annotation of the VaxxHesitancy set.
> We hope that if published, our paper will 1) provide a useful dataset resource for the research community, and 2) encourage the community to gather confidence scores during annotation and make them publicly available alongside aggregated labels.
>
> **3. Results analysis**
>
> Our results analysis was constrained by the 4 page short paper limit. If accepted, we will use the additional 1 page to further contextualise and discuss our results (including discussion of variance and experimental setup discussed below).
>
> **4. Notation**
>
> We agree the notation should be made clearer. The y hat we calculate from Eq 1 is used twice in Eq 2. Once as the prior in the numerator i.e. p(lim); once for calculating the marginal probability in the denominator i.e. p(am).
>
> **5. Experimental Setup**
>
> We do use portions of the test set for training. (For each round of cross-validation, we take 4  / 5 folds from the test set for training (along with the entire train set) and evaluate on the remaining fold)
>
> This is motivated by the fact that the original VaxxHesitancy train-test split distributed samples unevenly: low annotator agreement samples were placed in the train set and high-agreement samples in the test set. For generating soft labels, we want to train with a mixture of high-agreement and low-agreement annotations, so we include a portion of the original test set for training.
>
>
> **6. Numbers from the Appendix**
>
> We apologise for this mistake. Here we accidentally reference values from Table 3 (Appendix), which uses the 1-0.1 confidence scale. The main paper uses the 1-0.6 scale, so the values should be 71.1 and 70.4 (Table 1). However, the point demonstrated remains unchanged, as the trend is consistent across confidence scales. (In both cases, on the CDS dataset, soft labels perform better without calibration).
>
> **7.Cross-validation Variance**
>
> High variance between folds was due to the presence of ‘hard-to-classify’ samples in certain folds. These were samples where annotators relied on multimodal information to come to a decision (e.g. viewing a video embedded in the tweet).
>
> As expected, model performance was poor on these samples, leading to a lower overall f1 score for those folds. However, we used the same seed to generate folds across all experiments, so this variation between folds is consistent across different settings.
>
>
> **8. Hyperparameters**
>
> We did not perform a hyperparameter search for this study. Instead, we relied on our empirical knowledge to determine the hyperparameters. We appreciate the reviewer’s concern regarding this matter. Since this work is still in progress and has been submitted as a short paper, we will take this reviewer feedback into account for our subsequent experiments.

---

### Official Review · Reviewer_MMN4 · 2023-08-05

**Soundness:** 3

**Excitement:**

3: Ambivalent: It has merits (e.g., it reports state-of-the-art results, the idea is nice), but there are key weaknesses (e.g., it describes incremental work), and it can significantly benefit from another round of revision. However, I won't object to accepting it if my co-reviewers champion it.

**Paper Topic And Main Contributions:**

The authors introduce an important direction in the domain of modeling annotators. The paper proposes a novel method for generating enhanced soft labels that takes into account the variability in confidence levels among different annotators and the paper investigates how annotator confidence and secondary labels can be used to generate soft labels when the number of annotators is low.

**Questions For The Authors:**

A The authors have utilized only two datasets but there is a slew of other datasets which does fit the bill of the requirements, any reason as to why? Datasets such as Social Bias Inference Courpus/GoEMotions/Toxic Ratings.
B. How is the confidence calculated by the annotators? Is it a self rating?

**Reasons To Accept:**

1. The paper is a short paper of 4 pages and authors have definitely fit in lot of important context about their methods.
2. Their findings are stress tested with various baselines that are standard in the domain.

**Reasons To Reject:**

Included in questions for author to justify.

**Reproducibility:**

5: Could easily reproduce the results.

**Reviewer Confidence:**

4: Quite sure. I tried to check the important points carefully. It's unlikely, though conceivable, that I missed something that should affect my ratings.

---

> ### Author Rebuttal · Authors · 2023-08-28
>
> Thanks for your helpful questions and feedback!
>
> **B: Confidence scores**
>
> Confidence scores are self-reported by annotators, who were provided with definitions. For example, in VaxxHesitancy, a confidence of 3 is “Pretty confident about the annotation (I’m pretty sure about the annotation, but there might be in high chance that other annotators may label it in a different category.)”
>
> When generating soft labels, these are converted to values between 0 and 1, in the manner described in Line 279.
>
>
> **A: Datasets**
>
> In order to generate soft labels, we require data that contains individual annotator confidence scores. This was a surprisingly limiting restriction, and we hope that our paper will encourage researchers to gather and publish this information.
>
> For the datasets that you mention:
> - GoEmotions and Toxic Comment Classification do not offer annotators a way to express their confidence. (Toxic Comments does have a Rating set which evaluates toxicity based on score. However, this is only a test set, with no training data provided).
> - Social Bias Inference (SBIC) provides annotator confidence information. However, the range of possible confidences is too small to generate meaningful soft labels. (Most classes are only annotated with the options  ‘yes, maybe, no’ or ‘yes, no’. Only one class is annotated with the options ‘yes, probably, probably not, not’).
> - We conducted a preliminary experiment which confirms this. When trained on hard labels, BERT base uncased achieves  83.9 F1 on the SBIC ‘Intent’ class. However, training with soft labels results in a performance drop to 72.5, demonstrating that the annotation-style of this dataset is unsuitable.
>
> Additionally, while ‘soft label’ datasets do exist (such as ChaosNLI), these are created by aggregating annotator hard labels (and so individual annotator confidence is unavailable).

---

### Official Review · Reviewer_N6xf · 2023-08-11

**Typos Grammar Style And Presentation Improvements:** No
**Soundness:** 3

**Excitement:**

3: Ambivalent: It has merits (e.g., it reports state-of-the-art results, the idea is nice), but there are key weaknesses (e.g., it describes incremental work), and it can significantly benefit from another round of revision. However, I won't object to accepting it if my co-reviewers champion it.

**Missing References:**

No

**Paper Topic And Main Contributions:**

The paper presents a method for generating improved soft labels by leveraging consensus among annotators to align their confidence levels.
The main contributions are:
1. Authors utilize the Bayesian approach to align individual annotators' confidence scores.
2. Authors present an innovative dataset designed to support research focused on the application of soft labels in the field of Natural Language Processing.
3. Paper provides a solution to the issue of generating high-quality soft labels with limited annotator resources.

**Questions For The Authors:**

See weakness.

**Reasons To Accept:**

1. The paper challenges the conventional practice of discarding annotator disagreement and ambiguity in single-label classification tasks. By proposing the use of soft labels derived from these uncertain annotations, the approach maximizes the utilization of available data. This innovative method acknowledges that labeling ambiguity often stems from complex real-world situations and the lack of contextual cues in the data, making it crucial to extract as much information as possible from annotations.

2. The paper addresses a well-recognized limitation in data annotation and training methodologies for single-label classification tasks. By introducing a novel soft label method, the approach provides a practical solution to tackle the challenges posed by ambiguous labels.



**Reasons To Reject:**

1. Insufficient evidence or validation might undermine the credibility of the approach, raising concerns about the reliability of the results and conclusions. The authors did not provide any source to the dataset which might not be publically available.

2. The Methodology section could be made simpler and simpler by using simpler notations.

**Reproducibility:**

3: Could reproduce the results with some difficulty. The settings of parameters are underspecified or subjectively determined; the training/evaluation data are not widely available.

**Reviewer Confidence:**

3: Pretty sure, but there's a chance I missed something. Although I have a good feel for this area in general, I did not carefully check the paper's details, e.g., the math, experimental design, or novelty.

---

> ### Author Rebuttal · Authors · 2023-08-28
>
> Thank you for your comments and suggestions!
>
> 1.
> We will make our dataset publicly available alongside the paper so that researchers can replicate/build upon our results. The full dataset is currently provided in the supplementary material to maintain anonymity, however we will publish the dataset in an open data repository upon acceptance.
>
> 2.
> We agree that the Methodology section could be made simpler and will work towards improving notation for the camera-ready version.

---

### Meta-Review · Senior_Area_Chairs · 2023-10-04

**Recommendation:** 3

**Metareview:**

The paper introduces a method for improving soft labels by aligning annotator confidence levels using a Bayesian approach. It also presents a novel dataset for soft label research in NLP.

*Reasons To Accept:*
1. Challenges conventional practice by utilizing uncertain annotations for soft labels, maximizing data utilization.
2. Addresses a recognized limitation in data annotation for single-label classification.
3. Innovative dataset creation supports future research.

*Reasons To Reject:*
1. Lack of dataset source accessibility raises concerns about evidence and validation.
2. Methodology could benefit from simplification for better clarity.

**Summary:**

The paper introduces an innovative method for generating soft labels, addressing a significant issue in the field of data annotation for single-label classification tasks. It challenges conventional practices and provides valuable contributions. However, concerns regarding dataset accessibility, method complexity, and comparison against more recent baselines need attention. The reviewers' confidence levels vary, indicating some uncertainty, but overall, the paper shows promise and potential for acceptance with revisions.

---

### Decision · Program_Chairs · 2023-10-07

**Decision:**

Accept-Findings

**Comment:**

The paper introduces a method for improving soft labels by aligning annotator confidence levels using a Bayesian approach. It also presents a novel dataset for soft label research in NLP.

*Reasons To Accept:*
1. Challenges conventional practice by utilizing uncertain annotations for soft labels, maximizing data utilization.
2. Addresses a recognized limitation in data annotation for single-label classification.
3. Innovative dataset creation supports future research.

*Reasons To Reject:*
1. Lack of dataset source accessibility raises concerns about evidence and validation.
2. Methodology could benefit from simplification for better clarity.

**Summary:**

The paper introduces an innovative method for generating soft labels, addressing a significant issue in the field of data annotation for single-label classification tasks. It challenges conventional practices and provides valuable contributions. However, concerns regarding dataset accessibility, method complexity, and comparison against more recent baselines need attention. The reviewers' confidence levels vary, indicating some uncertainty, but overall, the paper shows promise and potential for acceptance with revisions.